# Comprehensive functional core microbiome comparison in genetically obese and lean hosts under the same environment

Marina Martínez-Álvaro [1✉], Agostina Zubiri-Gaitán [2], Pilar Hernández [2], Michael Greenacre [3], Alberto Ferrer[2] & Agustín Blasco [2✉]

Our study provides an exhaustive comparison of the microbiome core functionalities (captured by 3,936 microbial gene abundances) between hosts with divergent genotypes for intramuscular lipid deposition. After 10 generations of divergent selection for intramuscular fat in rabbits and 4.14 phenotypic standard deviations (SD) of selection response, we applied a combination of compositional and multivariate statistical techniques to identify 122 cecum microbial genes with differential abundances between the lines (ranging from −0.75 to +0.73 SD). This work elucidates that microbial biosynthesis lipopolysaccharides, peptidoglycans, lipoproteins, mucin components, and NADH reductases, amongst others, are influenced by the host genetic determination for lipid accretion in muscle. We also differentiated between host-genetically influenced microbial mechanisms regulating lipid deposition in body or intramuscular reservoirs, with only 28 out of 122 MGs commonly contributing to both. Importantly, the results of this study are of relevant interest for the efficient development of strategies fighting obesity.

[1] Scotland´s Rural College, Edinburgh, UK. [2] Universitat Politècnica de València, Valencia, Spain. [3] Universitat Pompeu Fabra, Barcelona, Spain. ✉email: marina.alvaro@sruc.ac.uk; ablasco@dca.upv.es

Obesity has markedly increased worldwide over the past 40 years, and projections indicate that global obesity prevalence will exceed 18% in men and 21% in women by 2025[1]. Overweight and obesity have medical consequences such as increased risk of diabetes, cardiovascular disease, cancer, or depression[2–4]. Therefore, obesity remains a major public health and also an economic concern. Dietary excesses and also host genetic factors are the main obesity causes in humans[5], with heritability estimates ranging from 0.40 to 0.70 in humans[6] and farm animals[7,8], and some loci already identified for obesity (e.g., MC4R, leptin, Fob, and FTO)[9–11] and obesity-related metabolic disorders (NOD$_{1,2}$[12], apolipoprotein E[13,14], and CD$_{14}$[15]). Numerous studies have demonstrated that microbiota is also a key contributor to obesity[16–23]. Bäckhed et al.[16] were the first to report a striking difference in body-fat content observed between germ-free mice and conventionally raised mice, with the latter having 40% higher body-fat content[16]. After that, several studies have established that microbial pathways can contribute to obesity by altering apetite[24], energy extraction from ingesta[17–23], and inflammation and immunity processes via an increased production of microbial-derived metabolites[12,18,21–23,25–30]. At the same time, the gut microbiome composition is partially determined by the genes of the host[31,32], as for example, host genes regulating the pH and redox conditions, motility of the intestinal tract, immune response, or the presence of receptors responding to metabolites with microbial origin[33–38]. Given the key role that microbiota plays in energy expenditure and inflammation processes, it is likely that genetically obese hosts carry genes defining a gut environment that favors the colonization of specific microbes or microbial functions associated with obesity. Although the role of microbiota in overall host adiposity is widely studied from a phenotypic point of view[39–41], many of these studies failed in disentangling which of these microbial mechanisms are under host genetic control. In their study, Turnbaugh et al.[42] investigated the effect of an obesity-related host gene (leptin) in the microbiome and showed its potential effect on the capacity of the microbiome to harvest energy from the diet. However, lipid deposition is a complex trait and its genetic background comprises a large set of loci with minor effects that may influence the gut microbiome. Until now, the effect of the complete genetic background for lipid deposition on the functional microbiome has not been studied, mainly due to the lack of appropriate experimental material under controlled environmental conditions. Our work attempts to cover this gap of knowledge, which is paramount for the development of obesity treatments (e.g., probiotics) in individuals with genetic determination to develop the condition.

Obesity comprises increased rates of lipid deposition and enlargement of fat cells[2] in body fat and also in intramuscular fat depots[43]. Whereas the role of the microbiome composition in the former has been extensively studied[16,17,26,44], published works about the microbiome influence on intramuscular fat deposition are restricted to phenotypic studies in livestock species[45,46]. During the last few years, our group has developed a successful divergent selection experiment for intramuscular fat content in Longissimus thoracis et lumborum (LTL) muscle in rabbits[7] with a response of 4.14 standard deviations (SD) after 10 generations (Supplementary Figure 1). Rabbits are a more appropriate experimental model than the most widely used mice for the study of lipid metabolism in humans, as their lipoprotein metabolism, cardiovascular system, and obesity-related clinical signs are more similar to those of human than the same systems of mice[47–49]. The divergent genetic lines have been contemporary raised and kept in the same environment during the whole experiment, exclusively differing in the genes underlying intramuscular fat content and genetically correlated traits. The main advantage of our experimental design, in comparison with other genetic studies, is that direct and correlated responses to selection are the observed phenotypic differences between lines at the same generation, independent of any genetic parameters or models. In previous studies, we demonstrated that the genetic background underlying intramuscular fat in LTL muscle influences intramuscular fat content of several muscles[50] and overall body adiposity (Supplementary Fig. 1 and Supplementary Table 1); lipogenic activity of muscular, adipose[51], and liver tissues[52]; liver weight[52], adipocyte hypertrophy[51], and C16:0 intramuscular content[53], all greater in the high in comparison with the low line, therefore referred from now as obese and lean lines. Our rigorous experimental design permits a comprehensive identification of the microbiome functionalities influenced by the host genetic background for lipid deposition by comparing the microbiome metabolism of the obese and lean lines. By using whole-metagenome sequencing of 89 cecal content samples, here we provide an extensive description of the functional core microbiome of rabbit cecum, until now limited to indirect inferences based on taxonomy[54], or to specific enzymatic activities[55]. We quantified the abundances of 3,937 microbial genes (MGs) present in all the samples (Supplementary Data 1), 3,395 of them classified, based on Clusters of Orthologous Groups of proteins database (COG)[56,57], in 23 functional modules (Supplementary Fig. 2) comprising 94.38% of the total abundance in the cecum. Our study revealed a strong correlated response to selection confirming that genetics of intramuscular fat content determines a gut environment that favors specific microbial core functions (i.e., MGs) associated with fat-mass accretion in muscle. These specific MGs were involved in the metabolism of bacterial-origin metabolites lipopolysaccharides, peptidoglycans, and lipoproteins, branched-chain fatty acids (BCFAs), propionate and acetate, bile acid transport, mucin biosynthesis, quorum sensing, or antibiotic resistance. In addition, we elucidated that a substantial part of the microbial mechanisms identified do not contribute to lipid deposition in body-fat depots, suggesting a certain independence between microbial metabolic pathways regulating fat deposition in different body sites.

The study of Gloor et al.[58] warned about the necessity of using log-ratio transformations when analyzing data from whole-genome sequencing as they are compositional, although proposed transformations (e.g., centered or isometric log ratio) difficult the result interpretation[59,60] or are not strictly subcompositionally coherent[61]. A further remark from our study is the application of an innovative strategy[62] to select the most adequate set of additive log ratios (alrs) in order to alleviate the spurious dependences among compositional microbial abundances, conserve the original isometry of the data, and facilitate its interpretation.

## Results and discussion
**The functional capacity of cecal core microbiome was modified as a response to selection for intramuscular fat content.** The main hypothesis of this study is that genetic determination of the polygenic trait intramuscular fat content comprises host gut traits that permanently alter cecal core microbiome functionalities captured as MG abundances. Studies based on phenotypic associations between microbiome and host lipid deposition[22,23,39] do not differentiate between genetic and environmental causes for the association. Prior to any statistical analysis, the 3,937 relative abundances of cecum MGs were transformed into 3,936 alrs with respect to the relative abundance of reference MG RP-S1. Our set of MG alrs of choice reproduces the exact log-ratio geometry between the samples (Supplementary Fig. 3), is subcompositionally coherent[61], and presents clear interpretation

advantages with respect to other log-ratio transformations[59,60] (see "Materials and methods" and Supplementary Fig. 3 for reference MG selection criteria). We used partial least squares discriminant analysis (PLS-DA)[63] for the divergent lines and standard partial least squares analysis (PLS)[64] for intramuscular fat content to elucidate which of the 3,936 MG *alrs* are influenced by host genes responsible for intramuscular fat content. The differences between our lines on the microbiome function (detected by PLS-DA) should be mainly genetic, but to reduce the risk of obtaining differences due to genetic drift, we corroborate the findings (in the Karl Popper sense) by performing the linear association of microbiome function with intramuscular fat content (detected by PLS). When MG *alrs* are found to be relevant in both analyses, it is likely that they should be associated with the genetics of the trait.

Our hypothesis was confirmed by finding a reduced set of microbial functionalities with exceptional discrimination abilities between the lines and substantial prediction abilities for intramuscular lipid deposition after an internal validation approach. We identified 240 MG *alrs* able to discriminate between obese and lean lines with a cross-validation misclassification rate of 5% (PLS-DA, Fig. 1a and Supplementary Data 2). We also found 230 MG *alrs* that together were able to explain 79% of intramuscular fat-content variability among the individuals in a cross-validation approach ($Q^2$) (PLS, Fig. 1b and Supplementary Data 3). Of the 240 and 230 MG *alrs* identified in PLS-DA and PLS models, 122 overlapped in both analyses (Fig. 1c and Supplementary Data 4), which summed a cumulative relative abundance of 1.40%. We estimated the magnitude of the correlated responses to selection on these 122 MG *alrs* by computing the difference between lines for each of their *alr*-transformed abundance and expressing them in units of SD (Supplementary Data 4 and Fig. 2). Using Bayesian techniques[65,66], we established the actual probability of the differences between lines in absolute value being higher than zero ($P_0$), which should not be confounded with a *P*-value. Of the 122 MG *alrs*, 86 evidenced differences with a $P_0 \geq 0.90$, 72 being more

abundant in the obese line. A lack of linearity between MG *alrs* and intramuscular fat content, or interactions between MGs, among other causes, may explain the lack of evidence when finding differences between lines in the remaining 26. Because of the almost constant log-transformed relative abundance of reference MG *RP-S1* occupying the *alr* denominator (its coefficient of variation is 2.3%), differences between lines in MG *alrs* can be strictly associated with different log-transformed abundances in the numerator. In the following, we take advantage of this assumption to discuss our findings.

Log-ratio transformations are required when analyzing compositional whole-genome sequencing data with multivariate covariance-based methods to avoid the spurious dependences among microbial abundances and with the trait of interest[58]. In this study, we further tested the importance of applying a compositional transformation by comparing our results with those obtained when repeating our DA-PLS and PLS, this time using MGs expressed in relative abundances (Supplementary Data 5, 6 and 7). While the discrimination and prediction abilities of the now MG-relative abundances selected were maintained (279 MGs showed a 6% cross-validation misclassification rate between the lines in PLS-DA, and 344 MGs explained a $Q^2 = 73\%$ of intramuscular fat-content variation); more than half of the 146 MGs overlapping between both analyses (79 out of 146) did not coincide with those 122 MGs obtained as numerators in the *alr* transformation. Our analysis supports that avoiding a compositional transformation when performing covariance-based analysis (e.g., PLS and PLS-DA) in MG databases of the gut microbiome may lead to spoiled results.

**The abundance of MGs involved in lipopolysaccharides, peptidoglycans, lipoproteins, and mucin components is increased in hosts genetically determined to accumulate high amounts of intramuscular fat.** The genetic selection process altered the abundance of 8 MGs involved in lipopolysaccharide biosynthesis (*lpxA, lpxB, lpxK, lpxH, waaF,* and *waaR*) and transport (*lptD* and *lptF*), from which *lptD, waaF,* and *waaR* presented strong

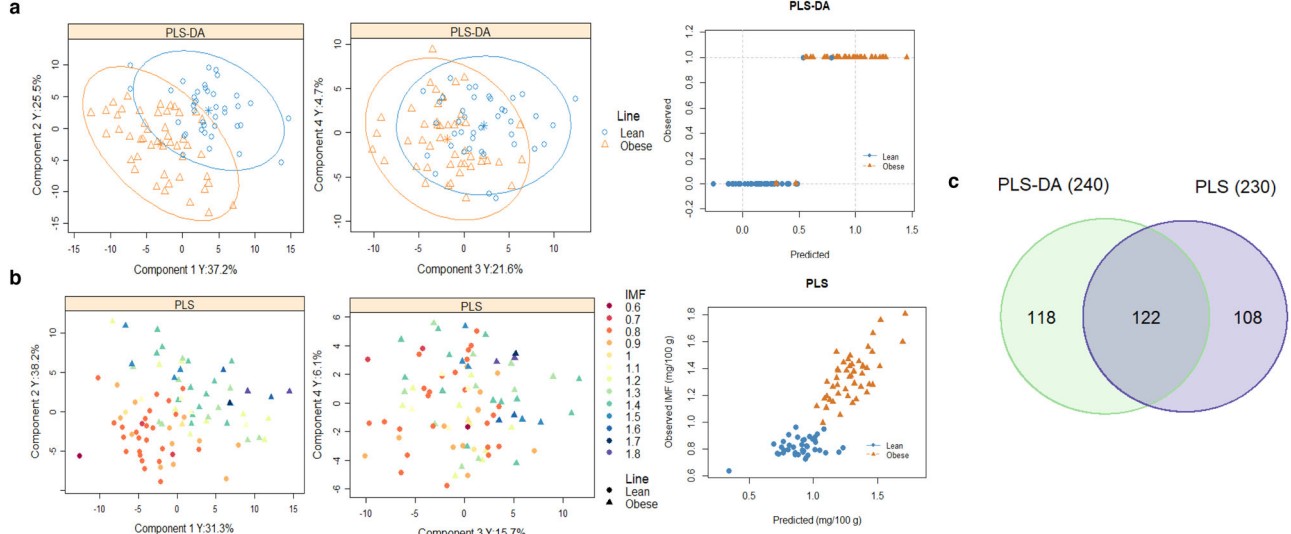

**Fig. 1 Comprehensive identification of cecal core microbial gene (MG) additive log-ratio (*alr*) transformed abundances evidencing a correlated response to selection for intramuscular fat content.** Individual plots ($n = 89$) and observed vs. cross-validation predicted values (with seven validation groups) obtained from **a**. Partial least squares discriminant analysis (PLS-DA) (each line is represented by a different color, contoured by 95% confidence ellipses) and **b**. Partial Least Squares (PLS) analysis (colored by increased intramuscular fat content, g/100 g of muscle (IMF)). The final models were built based on 4 latent components in both cases and 240 or 230 MG *alrs*, respectively. PLS-DA model showed a misclassification rate of 0%, and a cross-validation misclassification rate of 5%. PLS model showed a goodness of fit ($R^2$) of 91% and a goodness of prediction after cross-validation ($Q^2$) of 79%. **c** Venn diagram displays 122 MG *alrs* selected in both PLS-DA and PLS analysis, while 118 were selected exclusively in PLS-DA, and 108 exclusively in PLS model.

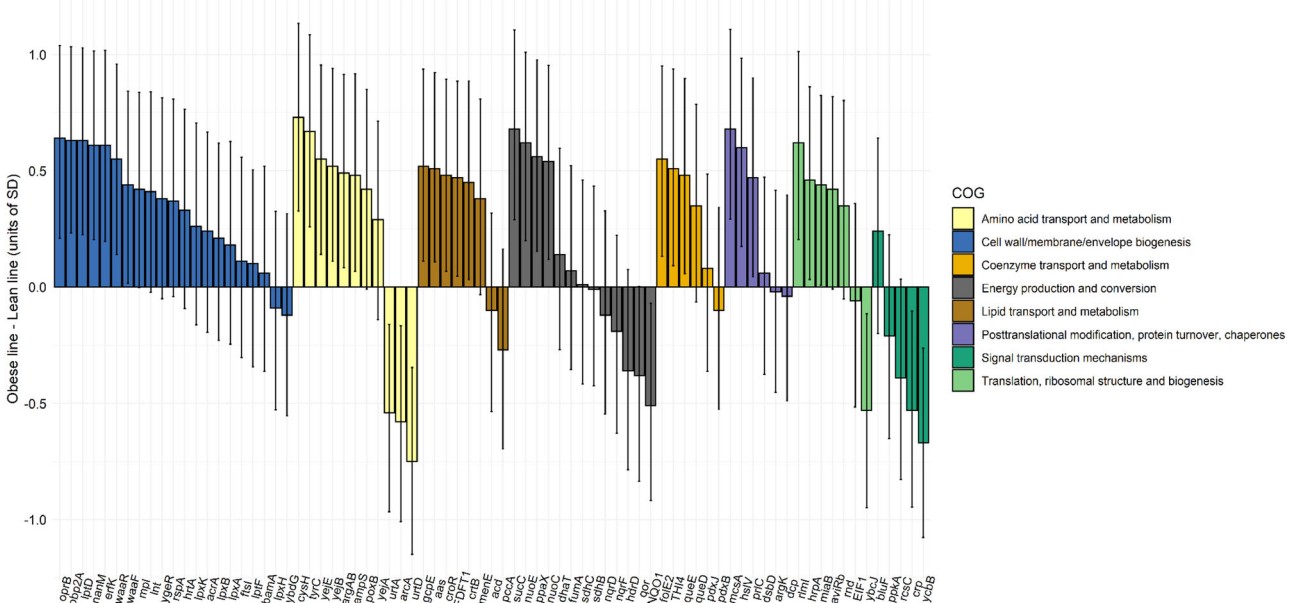

**Fig. 2 Differences between obese ($n = 47$) and lean ($n = 42$) lines in core microbial gene (MG) additive log-ratio (*alr*)-transformed abundances presenting a correlated response to selection for intramuscular fat content.** Bars represent differences in units of standard deviation (SD) of MG *alrs* and highest posterior-density intervals at 95% probability. Colors represent the classification of MGs according to Cluster of Orthologous Groups of proteins (COG). Full information of the MGs can be found in Supplementary Data 4. The 122 MG *alrs* commonly identified in partial least squares discriminant analysis and partial least squares analysis were classified into 20 different COGs, but only the most represented are displayed in the graph: cell wall/membrane/envelope biogenesis (21 MGs), energy production and conversion (13 MGs), amino acid metabolism (11 MGs), lipid transport and metabolism (8 MGs), translation, ribosomal structure, and biogenesis (7 MGs), coenzyme transport and metabolism (6 MGs), post-translational modification protein turnover, chaperones (6 MGs), and signal-transduction mechanisms (5 MGs).

evidence ($P_0 \geq 0.98$) of being more abundant in the obese line (differences between lines ranging from 0.43 to 0.63 SD, Fig. 2 and Supplementary Data 4). We also observed an increase in the abundance of 4 MGs involved in the biosynthesis of peptidoglycans in the obese line: *erfK*, *pbp2A*, and *mpl* (0.63–0.42 SD, $P_0 \geq 0.97$), and *ftsI* with lower evidence (0.12 SD, $P_0 = 0.71$). Bacterial lipopolysaccharides and peptidoglycans are microbiota-derived endotoxins mainly found in Gram-negative bacteria[30,67] contributing to metabolic endotoxemia[25,26] and fat-mass development[18,22,23,26] by boosting intestinal permeability[26,41,68,69] and triggering proinflammatory responses when binding $CD_{14}$, $TLR_4$[27] (lipopolysaccharides), and $NOD_1$ and $NOD_2$[12,28–30] (peptidoglycans) host receptors. Besides, obese line harbored an increased abundance of 2 MGs involved in the metabolism of lipoproteins: apolipoprotein N-acyltransferase *lnt* and lipoprotein *ygeR* (0.41 and −0.38 SD, $P_0 \geq 0.95$). Lipoproteins are universally distributed in bacteria[70] and play important roles, including nutrient uptake, transmembrane signal transduction, antibiotic resistance, and adhesion to host tissues[71]. They also function as trigger molecules for the activation of host innate immune response via $TLR_2$[72] host receptors, contributing to obesity-induced inflammation and insulin resistance[73,74]. Together, these results elucidate that microbiome-induced obesity via lipopolysaccharides, peptidoglycans, and lipoprotein metabolites has a host genetic component. The presence of $TLR_4$, $TLR_2$, NOD, and $CD_{14}$ receptors is regulated by host genes[12–15], for which we hypothesize could be a part of the host genetic background modified by the genetic selection process.

Another interesting result was MG *nanM* ($P_0 = 1.00$) involved in the synthesis of N-acetylneuraminic acid (Neu5Ac)[75]—the most common sialic acid of gut mucins in host mucosal surfaces —and *nahK* phosphorylating N-acetylglucosamine and N-acetylgalactosamine components of mucins[76] being greater in the obese line (0.61 and 0.38 SD, $P_0 \geq 0.95$). Sialic acid in mucous regions works as a host–microbiome cross-talk mechanism[76]

favoring the colonization of bacteria carrying sialic acid metabolism genes (e.g., mucin degrader *Akkermansia*)[77,78], which promote the release of host sialic acids[76]. In this study, we also identified which microbial taxa harbored the greatest number of the 122 MGs associated with host genes for intramuscular lipid deposition. Interestingly, the 10 top microbial taxa contained between 65 and 167 unique proteins mapping into 46–77 MGs (Fig. 3 and Supplementary Data 8) and among them, the single taxa annotated at genera level was *Akkermansia* and carried in its genome MGs involved in the biosynthesis of mucin components, lipopolysaccharides and peptidoglycans, and metabolism of lipoproteins (*nanM*, *nahK*, *lpxB*, *lpxK*, *waaR*, *lptF*, *lptD*, *pbp2A*, *ftsI*, *mpl*, and *lnt*), suggesting that *Akkermansia* could be playing a main role in genetically-induced obesity. However, a higher abundance of mucin degrader *Akkermansia*[79] residing in the mucous layer has been suggested to decrease obesity and type-2 diabetes incidence in mice[80] and alleviate metabolic syndrome[81] and obesity[82] in humans. A proper analysis quantifying the abundance of *Akkermansia* genera in the genetic rabbit lines and studying their correlated response to selection will be necessary to cast light on this point.

**Changes in microbial energy production and conversion pathways in cecum microbiome as a consequence of the selection for intramuscular fat content.** Three different types of bacterial NADH dehydrogenases are key in the ATP-generating process by transferring electrons from NADH to quinone, differing in their catalytic mechanisms and cofactors[83]. Our results show that cecum of obese individuals is enriched with $H^+$ pumping NADH:ubiquinone oxidoreductases (*nuoE* and *nuoC* are 0.62 and 0.54 SD more abundant in the obese line, $P_0 \geq 0.99$, Supplementary Data 4), while cecum of lean is enriched in nonpumping NAD(P)H dehydrogenase *NQO1* (-0.51, $P_0 = 0.99$) and $Na^+$ pumping NADH:quinone oxidoreductases (*nqrD* and

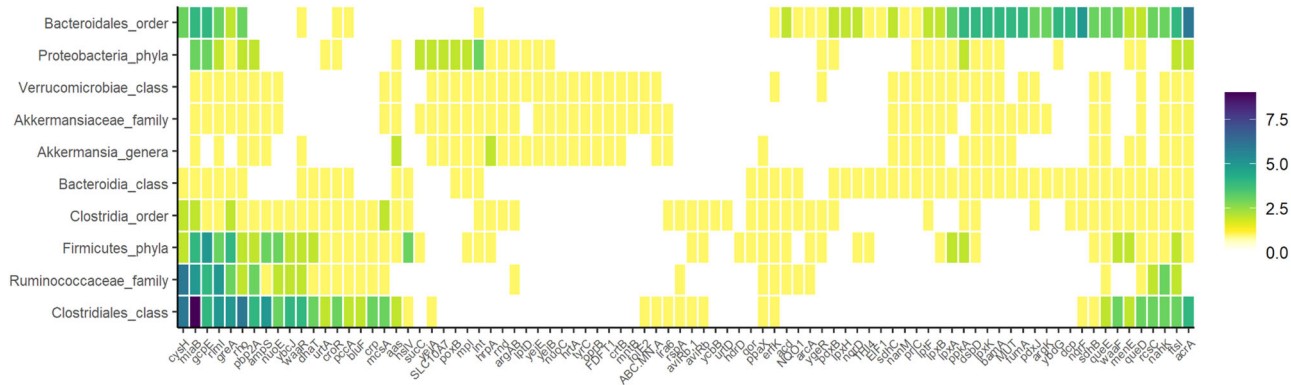

**Fig. 3 Top 10 microbial taxa highly enriched with unique proteins classified within KEGG Orthologue groups (or microbial genes) presenting a correlated response to selection for intramuscular fat content.** Color scale represents the number of unique proteins mapping into each KEGG Orthologue group (or microbial gene) identified within each taxa. Full names of microbial genes are given in Supplementary Data 4.

$nqrF$ −0.12 and −0.18 SD, $P_0 \geq 0.70$). While $nuo$ and $nqr$ generate electrochemical gradients ($H^+$ or $Na^+$), $NQO1$ does not contribute to the formation of a potential across the respiratory membrane[83]. We may speculate the existence of different flows of reducing equivalents in the cecum of obese vs. lean line as a result of the selection process, which leads to different bacterial adaptation in order to assure their energy conservation according to the gut environment. Moreover, obese line showed a higher abundance of citrate-cycle MG $sucC$ (0.68 SD, $P_0 = 1.00$) catalyzing the synthesis of succinyl-CoA from succinate, which has pro-inflammatory properties on lipopolysaccharide-activated macrophages and contributes to adipose-tissue inflammation[84]. $SdhB$ and $sdhC$ encoding different subunits of *succinate dehydrogenase/fumarate reductase* were also altered by the selection process, although their differential abundance between lines was not conclusive. In contrary, $hdrD$ involved in methane metabolism was more abundant in the lean line (−0.36 SD, $P_0 = 0.95$). Some studies observed a depletion of ubiquitous *Methanobrevibacter smithii* in feces of obese individuals[85,86], and a lower abundance of $H_2$-oxidizing methanogens in obese than lean pig breeds[87], explained by obese individuals minimizing carbon losses by diminishing their methane release. While different breeds may differ in many different host genes, our study suggest that this potential mechanism to reduce carbon losses is specifically associated with host genes for intramuscular fat content.

**Genetic selection for intramuscular fat alters the metabolism of lipids in cecum microbiome.** We found specific MGs involved in glycerophospholipid metabolism ($aas$), biosynthesis of carotenoids ($crtB$), terpenoid-backbone biosynthesis ($gcpE$), ubiquinone and other terpenoid–quinone biosynthesis ($menE$), steroid biosynthesis ($FDFT1$), and glycoxylate and dicarboxylate metabolism ($croR$) being more abundant in the obese line (from 0.36 to 0.52 SD, $P_0 \geq 0.96$). The abundance of $poxB$ encoding for an enzyme acting in the interconversion of pyruvate into acetate was also increased in the obese line (0.42 SD, $P_0 = 0.97$), acetate being a major substrate for *de novo* lipogenesis in rabbit's liver[88]. This result matches with the increased liver lipogenesis rate previously observed in the obese line[52], which was 1.51 SD greater than in the lean line for *glucose-6-phosphate dehydrogenase* activity. In contrast, $pccA$ involved in the degradation of branched-chain amino acids into BCFAs was more abundant in the lean line (−0.27 SD, $P_0 = 0.89$) and $acd$ and $MUT$ involved in the same pathway presented the same trend. Several studies noticed lower levels of BCFAs in the serum of excess-weight individuals than in the serum of lean[89,90], and this parameter increased linearly with individual's weight loss[86]. It is hypothesized that BCFAs are

negatively associated with obesity because they inhibit the liver synthesis of triglycerides[89,91]. Enzyme propionyl-CoA carboxylase ($pccA$) and $MUT$ play a key role in the synthesis of propionate, another inhibitor of *de novo* liver lipogenesis[92]. In addition, propionate[17] promotes leanness increasing energy expenditure and enhancing the production of satiety anorexigenic hormones inhibiting appetite via G-protein-coupled receptors (e.g., $GPR_{43}$ and $GPR_{41}$) expressed in gut epithelial cells and adipocytes[18–20]. In this line, host gene $RAPGEF1$ encoding a protein factor linked to G-protein-coupled receptors has been recently associated with microbiome composition in humans[37]. The putative influence of the microbial propionate synthesis on our animals' intake and therefore fat deposition, is supported by previous data from this experiment, which showed greater feed intake in the obese line[93].

**Selection for intramuscular fat alters the microbial metabolism and transport of amino acids.** Two MG abundances presenting a strong and positive correlated response to selection for high host lipid deposition: $cysH$ catalyzing the reduction of activated sulfate in sulfur metabolism (0.73 SD and $P_0 = 1.00$); and $tyrC$ in phenylalanine, tyrosine, and tryptophan metabolism (0.65 SD, $P_0 = 1.00$). Sulfate-reduction pathway and aromatic amino-acid metabolites (i.e., tyrosine and phenylalanine) have been strongly associated with type-2 diabetic individuals[94] or with the development of future diabetes[95]. Arginine metabolism was another microbial metabolic pathway affected by the selection process as exhibited by $arcA$, $argAB$, and $ampS$ MGs ($P_0 \geq 0.99$). While $arcA$ catalyzing the hydration of L-arginine into L-citruline was more abundant in the lean line (−0.59 SD), $argAB$ and aminopeptidase $ampS$, which exhibits a high specificity toward peptides possessing arginine were more abundant in the obese line (both MGs having a difference between lines of 0.48 SD), elucidating a complex relationship among arginine metabolism and lipid deposition. On the other hand, two urea ATP-binding cassette (ABC) transporters $urtA$ and $urtD$ were more abundant in the lean line with differences of −0.54 and −0.75 ($P_0 \geq 0.99$)[96]. MGs encoding urea ATP- dependent uptake system have been widely found in the genome of members from Cyanobacteria phylum[97], which are negatively associated with abdomen-fat weight in pigs[98], although these MGs are only expressed under nitrogen-limiting conditions[99], otherwise passive diffusion is preferred.

**The genetics of intramuscular fat deposition are associated with microbial antibiotic resistance.** Obese line showed more abundance of $yejA$, $yejE$, and $yejB$, which encode ATP-binding cassette transporters of antibiotics (differences = 0.29, 0.55, and

0.51 SD, $P_0 \geq 0.91$); *aviRb* conferring bacteria antibiotic resistance (0.42 SD, $P_0 = 0.98$); and previously mentioned *pbp2A* and *ftsI* that are involved in the biosynthesis of the peptidoglycans interacting selectively and noncovalently with penicillin. Recent studies show that diet-induced obesity leads to greater bactericidal antibiotic resistance[40], and that antibiotics incites obesity by favoring the predominance of microbes with greater capacity of extracting energy from the diet[100]. Given that antibiotic effects on gut microbiota are host dependent[101,102], our results reveal that the antibiotic resistance–obesity association could be also caused by a host genetic component, independently of the diet.

**Other microbial pathways linked to host genes regulating intramuscular fat deposition.** Genetic selection to increase intramuscular fat content augmented the abundance of further MGs in folate biosynthesis (*folE2*, *queE*, and *queD* with differences between lines ranging from 0.35 to 0.55 SD, $P_0 \geq 0.95$), thiamine metabolism (*THI4* 0.51 SD, $P_0 = 0.99$), fructose and mannose metabolism (*lra6*, 0.72 SD, $P_0 = 0.97$), manganese transport (*mntB* and *ABC.MN.A*, 0.33 and 0.57 SD, $P_0 \geq 0.93$), and *SLC10A7* encoding for a bile acid importer (0.46 SD, $P_0 = 0.99$, Supplementary Data 4). Bile acids metabolized by microbiota are known as key contributors to host energy-homeostasis regulation and obesity[21] by promoting[103] or impairing[104] carbohydrate metabolism when binding to cellular receptors in various organs of the body (e.g., G-coupled receptor TGR5[103] and farnesoid X receptor[104]). Another important finding was that the obese line displayed an increasing abundance of 2 MGs in transcription (*greA* and *rho* from 0.53 to 0.56, $P_0 \geq 0.99$) and 5 MGs in translation, ribosomal structure, and biogenesis (*rlmI*, *hrpA*, *miaB*, *aviRb*, and *rnd*, from 0.36 to 0.62 SD, $P_0 \geq 0.95$). Given that protein synthesis is highly coupled with cellular growth[105], our results may indicate higher bacterial burden in genetically obese individuals, which has been previously observed in diet-induced obese mice[40]. In addition, the fact that these 7 MGs are extensively found in the genome of bacteria from Firmicutes phyla (Fig. 3 and Supplementary Data 8) could indicate an increased growth of Firmicutes in the cecum of the obese line. Finally, our results reinforce the hypothesis of a reduced interconnectivity within the microbiota in obese individuals[106] and reveal that this association is influenced by obesity host genes by displaying a greater abundance of MGs involved in biofilm formation (*ppkA*, *rcsC*, and *ycbB*) and quorum sensing (*crp*) in the lean line (differences ranging from −0.21 to −0.67 SD, $P_0 \geq 0.96$).

**Different microbial functional mechanisms influence lipid deposition in muscle and in main body depots.** The correlated response to selection for intramuscular fat in body-fat percentage equivalent to 1.5 SD ($P_0 = 1.00$, Supplementary Table 1 and Supplementary Fig. 1) demonstrates that both traits are partially regulated by common host genes. After showing that host genes for intramuscular fat content alter specific metabolic routes of the microbiome, a question still-to-solve is whether these microbial pathways also affect body-fat content. To this end, we first applied PLS to identify 184 MG *alrs* (Supplementary Data 9), which explained a substantial proportion of body-fat percentage variability ($Q^2 = 72\%$). Then, we compared these 184 with the 122 MG *alrs* presenting a correlated response to selection for intramuscular fat content and found that only 28 common in both analyses played an important role in the lipid-deposition rate of muscle and body depots. This result was corroborated by testing the ability of the 156 body-fat percentage-specific MG *alrs* to predict intramuscular fat content, and of the 94 intramuscular fat content-specific MG *alrs* to predict body-fat percentage, not

obtaining any PLS component with prediction ability in any case. While the microbial mechanisms commonly associated with intramuscular and body-fat percentage included, e.g., lipopolysaccharide and peptidoglycan biosynthesis (*waaR*, *lpxA*, *lpxB*, *lptD*, *ftsI*, and *erfK*), NADH dehydrogenases (*nqrD* and *qor*), methane metabolism (*hdrD*), or branched-chain amino acid degradation (*acd* and *MUT*), the microbiome function specifically driving lipid deposition in muscle comprises the biosynthesis of mucin components (*nanM* and *nahK*), metabolism of lipoprotein metabolism (*lnt* and *ygeR*), arginine metabolism (*argAB*, *ampS*, and *arcA*), and transport of bile acids (*SLC10A7*), among others (Supplementary Data 9). Taken together, these results suggest a substantial differentiation between host genetically influenced microbial mechanisms regulating lipid deposition in intramuscular or body-fat reservoirs.

## Conclusion

Our work casts light on the influence of the comprehensive genetic background for intramuscular lipid deposition on the cecal functional microbiome by using metagenomic data from two rabbit lines divergently selected for intramuscular fat content during 10 generations and reared under the same environmental conditions. We revealed a strong correlated response to selection on the functional microbiome composition of cecum, confirming that genetics of intramuscular fat content determines a gut environment that favors specific microbial functions associated with fat-mass accretion in muscle. Among the microbial functions enhanced in the obese line, we identified the metabolism of bacterial-origin metabolites lipopolysaccharides, peptidoglycans, and lipoproteins, known to bind specific host receptors regulated by host genes ($TLR_4$, $TLR_2$, NOD, and $CD_{14}$) to trigger inflammatory responses. Other microbial functions identified in this study (e.g., metabolism of BCFAs, propionate and acetate, or bile acid transport) are known to regulate liver lipogenesis but also appetite and carbohydrate metabolism in the host. The ATP-generating strategy preferred by microbes, which is essential for their colonization success, also differed between the lines, possibly because of their adaptation to different electrochemical gradients. A greater bacterial load is expected in the cecum of obese individuals based on greater abundance of MGs involved in protein synthesis and antibiotic resistance. Our study also revealed that host—microbiome interaction mediated by epithelial mucins and interspecies interactions (e.g., quorum sensing), which are of particular importance in energy regulation, is influenced by intramuscular fat genes. Importantly, our study confirms that all the different microbial functionalities identified concomitantly differ in the cecum microbiome of hosts with divergent genetic determination for fat deposition in muscle, which may be of relevant interest for the efficient development of strategies fighting obesity. In addition, we elucidated that a substantial part of the microbial mechanisms identified do not contribute to lipid deposition in body-fat depots, suggesting a certain independence between microbial metabolic pathways regulating fat deposition in different body sites.

## Methods

**Animals.** All experimental procedures involving animals were approved by the Universitat Politècnica de València Research Ethics Committee, according to council directive 2010/63/EU (European Commission Directive, 2010). A divergent selection experiment for intramuscular fat content in rabbits was performed during 10 generations. The base population that composed of 13 males and 83 females came from a synthetic line. From this base population, two divergent lines were selected for high (obese-line) and low (lean-line) intramuscular fat, each composed by 10 males and 60 females per generation. Two full sibs of the first parity of each doe were slaughtered at 9 wk of age and their intramuscular fat content in *LTL* was measured. All dams were ranked according to the average of the two phenotypic intramuscular fat values obtained from their offspring and the 20% best dams

provided all females for the next generation. Each sire was mated with five does, and only one male progeny of the sire from the highest-ranked mate was selected for breeding the next generation to reduce inbreeding. More details of this experiment can be found in Martínez-Álvaro et al.[7]. Data for this study were obtained from 89 animals randomly sampled from the tenth generation of selection, 47 from the obese line and 42 from the lean line. Sampling took place along three months during the winter (December 2017–March 2018). During this period, the farm temperature fluctuated between T°$_{min}$ 11.9 ± 1.1 and T°$_{max}$ 21 ± 2.5 °C. The farm (located at the Universitat Politècnica de València) has insulated roof and walls, controlled lightening and ventilation, and a cooling system. Cross-fostering was not practised. Kits were weaned at 28 days of age and then reared jointly within litters in common cages until slaughter. During fattening period, animals were fed *ad libitum* with the same commercial diet with an average composition of 16.0% crude protein, 16.5% crude fiber, 2.4% oil and crude fat, 7.6% ashes, 0.80% calcium, 0.60% phosphorus, and 0.26% of sodium, supplemented with vitamin A (10000 UI/KG), vitamin D3 (900 UI/kg), vitamin E (25 mg/kg), iron (78 mg/kg), cobalt carbonate (0.30 mg/kg), manganese (20 mg/kg), zinc (50 mg/kg), selenium (0.05 mg/kg), potassium iodide (1.0 mg/kg), copper (8 mg/kg), and bacitracin zinc antibiotic (100 ppm). The last seven days before slaughter at 9 wk of age, bacitracin zinc antibiotic was removed from the diet.

**Lipid-content measurements and cecal sample collection.** Animals were slaughtered after 4 h of fastening by exsanguination, prior electric stunning. Cecal-content samples were collected in the slaughterhouse, immediately after slaughter. The intestinal tract was removed from the abdominal cavity; cecum content was collected in 50-mL sterile Falcon tubes, homogenized, and aliquoted in 2-mL cryogenic tubes, obtaining samples that are representative of both solid and liquid phases. The aliquots were immediately submerged in liquid nitrogen and then stored at −80 °C, until the DNA extraction. After slaughter, carcasses were chilled for 24 h at 4 °C. The reference carcass weight, defined by the World Rabbit Science Association as the weight of the carcass without the head, liver, lungs, thymus, esophagus heart, and kidneys[109], was registered. The two main body-fat reservoirs in rabbit scapular and perirenal fat depots were excised and weighted. Body-fat content was estimated as the sum of their weights divided by the reference carcass weight. Muscle *LTL* was excised from chilled carcasses, ground, freeze-dried, and scanned by near-infrared spectroscopy (model 5000, FOSS NIRSystems INC., Hilleroed, Denmark) to measure intramuscular fat content (g of lipids/100 g of muscle in a fresh basis) applying the calibration equation previously developed by Zomeño et al.[110].

**Microbial gene-abundance measurements.** Bacterial genomic DNA was isolated from the frozen cecal samples using the DNeasy PowerSoil Kit (QIAGEN Inc, Hilden, Germany) using the manufacturer's instruction with the following modifications. Cecal samples (0.1 g) were disrupted with three glass beads in a bead homogenizer (BeadMill 4, ThermoFisher) in the presence of buffer C1, and incubated at 95 °C for 5 min. These steps were repeated twice. Sample tubes were spun at 10,000 g for 30 s and the supernatant was transferred to a new tube to follow with the manufacturer's instructions. In the final step, the DNA was eluted from the column in a volume of 100 μl. DNA concentration and purity were estimated first by spectrometry in a Nanodrop ND-1000 and by fluorometry in a Qubit 4 Fluorometer (Invitrogen, Thermo Fisher Scientific, Carlsbad, CA, USA) with the DNA dsDNA HS Assay kit (Invitrogen). Thirty-three microbial DNA samples (16 from obese and 17 from lean line) were sequenced at the facilities of the Sequencing and Bioinformatic Service of FISABIO (Valencia, Spain), while the remaining 56 cecal samples (31 from the obese and 25 from the lean line) were sequenced in Sistemas Genómicos (Valencia, Spain). One ng of each sample (0.2 ng/μl) was used for the shotgun library preparation, using the Nextera XT DNA Library Preparation kit (Illumina, Inc., San Diego, CA, USA) or the SureSelectQXT library preparation kit (Agilent Technologies, Inc., Santa Clara, CA, USA) for samples sequenced by FISABIO or Sistemas Genómicos, respectively, in both cases following the manufacturer protocol. Samples were sequenced by NextSeq 500 (FISABIO) or NextSeq 550 (Sistemas Genómicos) Illumina with 150 bp paired-end chemistry, with an average coverage of 4 million pair reads per sample with a minimum of 2 million. Raw sequencing data obtained were pre-processed to remove the reads that belong to the genome of the host by mapping them against the rabbit reference genome (OryCun v2.0.101) using the Bowtie2 software[106]. The bioinformatic analysis on the clean reads was performed using the SqueezeMeta[107] automatic pipeline. The first step consisted in the assembly of the reads to obtain longer contigs using the coassembly mode of the SequeezeMeta. This strategy performs the assembly pooling the reads from all the samples, which will result in longer contigs that can give better annotations. The second step consisted in the prediction of the open-reading frames (ORFs) inside the contigs using Prodigal[111]. The functional annotation of each ORF was then obtained by homology searching with sequences in the Kyoto Encyclopedia of Genes and Genomes database[108] using Diamond program[112]. The defined e-value and identity parameters for the homology search were $1 \times 10^{-3}$ and 50%, respectively. No hits below those parameters were considered. The functional ID of the highest-scoring hit was finally assigned to the ORF (best-hit approach of the SqueezeMeta pipeline). The last step consisted in the abundance estimation of the annotated ORFs (referenced as MG) in each sample, which was done by mapping

the individual reads against the corresponding MGs, using the Bowtie2 software[106]. We identified a total of 4,726 MGs in both groups. Among those, we kept 3,937 MGs present in all the samples equivalent to 99.7% of the total number of counts initially identified in the database. Biological information about each MG based on KEGG and Clusters of Orthologous Groups of proteins (COGs)[56] databases were obtained by KEGGREST[113] R package. To describe the cecal MG composition in rabbits, we estimated the relative abundance of each MG within each sample, i.e., its counts divided by the total sum of counts within a sample, also referenced as relative abundances. We used COG database to classify 3,395 out of the 3,937 MGs into 23 functional modules. To simplify their classification, we used only the first classification module when a MG was assigned to more than one module.

**Statistics and reproducibility**
*Compositional transformation of metagenomic data.* MG counts from high-throughput sequencing studies are not independent among one another because the sequencing instruments can deliver reads only up to the capacity of the instrument[58]. These constrained databases, referred to as compositional data, exhibit spurious correlations among the microbial variables and different correlation structure than the underlying count data in the original sample[58]. Normalizing the microbial abundances by the total sum of counts in a sample (relative abundances) does not solve the compositional problem as spurious correlations still remain. However, the log-ratio transformation of the microbial abundances relative to other features in the dataset provides information directly relatable to the original sample[58]. We applied an additive log-ratio[61] transformation (*alr*) of the MGs before covariance-based and differential abundance analysis. Additive log-ratio transformation permits comparison with other studies using different numbers of MGs in the total database (i.e., it has subcompositional coherence[61]) and it has a simpler interpretation in practice than other log-ratio transformations (e.g., centered or isometric), since they do not involve geometrical means[59,60]. Assuming *J* denotes the number of variables in the database (*J* = 3,937), the abundance of each MG within a sample was expressed as[114]

$$\ln\left(\frac{x_j}{x_{ref}}\right) = \ln(x_j) - \ln(x_{ref}), \quad j = 1, \ldots, J - 1, \, j \neq \text{ref}$$

where $x_j$ is the relative abundance of the *j*-th MG and $x_{ref}$ is the relative abundance of a reference MG. The housekeeping MG *RP-S1*, involved in ribosomal biosynthesis, was selected as the reference based on (1) almost perfectly reproducing the geometry of the full set of ratios in a Procrustes analysis[115] (Procrustes correlation with the whole set of log ratios is 0.9988, see Supplementary Fig. 3), (2) having very low variance in its log-transformed relative abundance (variance = 0.0018, see Supplementary Fig. 3, coefficient of variation = 2.13%, five-point distribution: min = − 2.13, 1$^{st}$ quartile = − 2.02, median = − 1.99, 3$^{rd}$ quartile = − 1.97, max = − 1.813), so the main variation of the *alr* comes from the numerator, (3) its relative abundance being uncorrelated with intramuscular fat content (Pearson correlation = − 0.07 ± 0.11), and (4) being highly abundant (mean relative abundance = 0.14%), which is generally associated with lower instrumental error[116]. To test the outperforming of the compositional transformation using additive log ratios in comparison with a noncompositional approach, we also run the same analyses with MGs expressed in relative abundances.

*Estimation of correlated responses to selection for intramuscular fat deposition on lipid traits and MG abundances.* Direct response to selection on intramuscular fat content and correlated responses in scapular, perirenal fat weight, and body-fat percentage were estimated as the phenotypic differences between obese and lean lines in a linear model, including line as a fixed effect, and solved with "Rabbit" software developed by the Institute for Animal Science and Technology (Valencia, Spain). The identification of the MG abundances in the cecum showing a correlated response to selection was evaluated by PLS-DA and PLS analysis[64] computed by SIMCA software (Umetrics, Umeå, Sweden), and R package mixOmics[117] was used to generate the graphs. Briefly, PLS and PLS-DA models are built based on latent components that maximize the covariance between the explanatory variables (MG *alrs*) and the traits of interest, in this case intramuscular fat in PLS and a dummy variable coding the obese/lean classification in PLS-DA. First, we detected discriminant MG *alrs* between lines by fitting a PLS-DA model with the 3,936 MG *alrs* as explanatory variables. The number of components in the model was selected based on a internal cross-validation procedure with seven cross-validation groups. One animal from the lean line was removed from the analysis for being detected as an outlier based on the distance of each individual to the model (DMOD criterion). After some exploratory analysis, MG *alrs* were selected if their variable importance for projection (VIP) > 0.8, or the jackknife 95% confidence interval of their regression coefficients did not contain zero. We followed an iterative process and stopped when the removal of one more MG *alr* decreased the model-prediction ability. The misclassification rate obtained by fitting the final model with the complete dataset (R²) and with 6/7 of the samples and classifying the remaining 1/7 (repeated for each cross-validation group, Q²) was computed. Second, we detected the more informative MG *alrs* explaining the variation within animals in intramuscular fat content by fitting a PLS model following the same iterative process as in PLS-DA checking the prediction ability of the model before (R²) and after the same cross-validation previously described (Q²). A permutation test was

used to check the robustness of PLS-DA and PLS models by obtaining a $P$-value $\leq 0.001$. Third, we compare the results in PLS-DA and PLS models and consider the MG *alrs* selected in both approaches to be the ones presenting a correlated response to selection.

The sign and the magnitude of the correlated response to selection on the abundance of each MG *alr* selected were studied by estimating the difference between obese and lean genetic lines in a linear model only including line effect with two levels. Linear models were solved by Bayesian procedures[65] using the Rabbit program developed by our institute[7] and assuming flat priors for all unknowns. To describe the marginal posterior distributions of the differences between lines, we estimated their median, their highest posterior-density interval at 95% probability, and the probability of the difference being higher or lower than 0 if the difference is positive or negative, respectively ($P_0$). We expressed the differences between lines in MG *alrs* as units of their SD.

*Identification of microbial gut mechanisms differentiating lipid deposition in muscle or in body-fat depots.* One of the objectives of this study was to search for MGs exclusively influencing intramuscular fat content independently of body-fat depot weight. We first fitted body-fat percentage in a PLS model following the same procedure as the one previously described. Then, we studied the overlap between the MG *alrs* explaining body-fat percentage with those showing a correlated response to selection in intramuscular fat content (overlapping between previously described muscle lipid content PLS and PLS-DA analysis) and considered as specific for intramuscular or body-fat content those MG *alrs* selected exclusively in one of the approaches. For further validation, we used MG *alrs* specifics for intramuscular fat to predict body-fat percentage, and in the same way, MG *alrs* specific for body-fat percentage to predict intramuscular fat expecting a negligible prediction ability.

**Reporting summary**. Further information on research design is available in the Nature Research Reporting Summary linked to this article.

## Data availability

Metagenomic sequence reads for all rumen samples are available under European Nucleotide Archive (ENA) under accession project PRJEB46755. Resolved metagenomics (microbial gene abundances) are available under request.

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

## Acknowledgements

We acknowledge Prof. J.J. Egozcue and V. Pawlosky for their advice in the analysis of compositional data. This work was supported by project AGL2017-86083-C2-1-P from the Spanish National research plan. M. Martínez-Álvaro acknowledges a post-doctoral grant (APOSTD/2017/060) from Generalitat Valenciana.

## Author contributions

A.B. designed the experiment. P.H. and A.Z collected the samples and performed the lab analyses. M.M., A.F., M.G. and A.B designed the statistical analyses and M.M. performed the statistical analyses. M.M., P.H., A.B. and A.Z. discussed the results. M.M. wrote the paper and all coauthors revised the paper.

## Competing interests

The authors declare no competing interests.
