## [Transparent Peer Review File · Communications Biology]

Reviewers' comments:

Reviewer #1 (Remarks to the Author):

In this manuscript, the authors used 89 rabbits with divergent genotypes for intramuscular fat content to investigate the effect of selection for IMF on the functional capacities of cecum microbiome. They identified 122 microbial genes (MGs) with differential abundances between the lines.

the major comments are listed in the following:

1. All the results about the associations of the changes of MGs abundances in cecum lumen with IMF were inferred from association study. There were no any experiments to confirm these hypotheses.
2. The analyses in this manuscript only focused on microbial genes. It should be interesting that the authors can link the microbial genes affected by selection for IMF to their corresponding bacterial species, and then, discussed the effects of these bacterial species on host IMF and the possible mechanism of these bacterial species influencing IMF through MGs.
3. The authors should provide the evidences that the genotypes associated with IMF had already fixed in two divergent lines, but not still be segregated.
4. line 48: ".....are restricted to phenotypic studies.....", However, phenotypic values are significantly correlated with genetics.
5. Line 81-88: this paragraph should be moved to the discussion section of methods section.
6. line 90: in this manuscript, the authors did not perform the metabolomics analysis, so it should be the functional capacities of cecum microbiome based on metagenomic sequencing data.
7. In my opinion, the authors should measure the concentrations of serum LPS, peptidoglycans, N-acetylneuraminic acid, lipoproteins, and other metabolites that the authors emphatically discussed in the result section of the manuscript. It would help to confirm that the changes of cecum microbiome functional capacities indeed led to the shifts of the corresponding metabolites, and these metabolites finally resulted in the different IMF contents.
8. Line 250-254: However, many studies have suggested that Akkermansia spp. can alleviate metabolic syndrome features in obese human. e.g., 10.1038/s41591-019-0495-2; 10.1038/s41591-019-0516-1.
9. line 369-370: Many studies in other mammals (such as pigs) have suggested different genetics backgrounds for IMF and body fat, e.g., backfat.

Reviewer #2 (Remarks to the Author):

The manuscript entitled "Comprehensive comparison of the cecum microbiome functional core in genetically obese and lean hosts under the same environmental conditions" is interesting, but the current form is not acceptable, it needs to be improved by addressing several below issues:

1. From what I know, there are some studies in the world which research about it. So, is this new topic? Authors should highlight more the reason why they did this work.
2. In the Main text, the consequences of obesity should be more clarified which will support the reasons of studies on obesity in general and on this topic in specific, several importance references you should read such as: (1). Medical Consequences of Obesity, The Journal of Clinical Endocrinology & Metabolism, Volume 89, Issue 6, 1 June 2004; (2), An update on obesity: Mental consequences and psychological interventions, Diabetes Metab Syndr. 2019 Jan-Feb;13(1):155-160; and (3).An update on physical health and economic consequences of overweight and obesity. Diabetes Metab Syndr. 2018 Nov;12(6):1095-1100.
3. In the second main text paragraph, you mentioned that "Rabbits are a more appropriate experimental model than the most widely used mice for the study of lipid metabolism in humans, as their lipoprotein metabolism, cardiovascular system and obesity-related clinical signs are more similar to those of human than the same systems of mice ". As far as I know, in previous studies, researchers still used mice as their research subject. Why is there a difference in the study subjects between the studies?
4. In the main text and introductory paragraph, you should give some information about the

microbiome in order to see overview of the whole article.

5. About the layout of the article, I find it not reasonable. If I were you, I would write the research method first, then the results and discussion.

6. In the Animal method, you mentioned in detail about the research subject, rearing process and amount of food. But, I haven't seen you mention farming conditions. such as laboratory conditions, etc..

7. In the Animal method, you mentioned that "A divergent selection experiment for intramuscular fat content in rabbits was performed during 10 generations". Why do you use 10 generations of mice for research?

8. In the method, in a study on the same topic, they used SSU rRNA gene amplification and pyrosequencing methods to evaluate the influence of host genetics on bacterial community composition (Campbell, J., Foster, C., Vishnivetskaya), T. et al, 2012), but your study does not cover this approach. Why?

9. How did you classified which rabbits were obese and which were not?

10. In the Microbial gene abundance measurements method, there are quite a few techniques applied, you should name each technique and talk in more detail. From there, readers can understand more easily

11. In the result, you have only concluded about the function of the cecum microbiome and their mechanisms, but the main objective of the study is the comparison of the cecum microbiome functional core in genetic obese and lean hosts under the same environmental conditions, you haven't mentioned it yet

12. One of the most important thing is that the molecular mechanisms underlying the interactions of the cecum microbiome and the hosts observed in this studies have not been enough studied well here, or the ways authors presented and interpreted the data were not so good that makes the readers difficultly to get the right information

13. In this research, you just stopped at making the conclusions of the study, stating some scientific bases and conclusions of previous studies. But you have not mentioned the shortcomings and applicability of the research.

REFERENCE

1. Campbell JH, Foster CM, Vishnivetskaya T, Campbell AG, Yang ZK, Wymore A, et al. Host genetic and environmental effects on mouse intestinal microbiota. *The ISME Journal*. 2012;6(11):2033-44.
2. Nguyen TLA, Vieira-Silva S, Liston A, Raes J. How informative is the mouse for human gut microbiota research? *Disease Models & Mechanisms*. 2015;8(1):1-16.
3. Turnbaugh PJ, Ley RE, Mahowald MA, Magrini V, Mardis ER, Gordon JI. An obesity-associated gut microbiome with increased capacity for energy harvest. *Nature*. 2006;444(7122):1027-31.

Reviewer #3 (Remarks to the Author):

Major Claims: Authors claim that their work elucidates that microbial biosynthesis of LPS, peptidoglycans, lipoproteins, mucin components, and NADH reductases are influenced by host genetic determination for lipid accretion in muscles. The authors also discuss the criterion to select adequate sets of additive log-ratios for compositional analysis. This emphasizes the importance of compositional transformation when performing covariance-based analysis. These claims are novel and would be of interest to the community.

Statistical Analysis: Authors wrote a very detailed description of the statistical analyses completed for the project. All statistical measures appear to be appropriate and reproducible.

1.Line 32-35= clarification is needed, do authors mean that the gut microbiome is determined by host genes along with other factors? If so, the sentence could be rephrased.

- 2.Lines 45-46; the definition of "obesity" and its role in lipid deposition could be clear.
- 3.Can the authors please explain why the cecal contents were examined?
- 4.Were fecal contents in the small intestine or colon examined? Why not pellets?
- 5.Lines 91-93; Hypothesis needs to be moved to the introduction. Also, the sentence structure makes the hypothesis difficult to read. Maybe format as the following, "The main hypothesis of this study is that genetic determination of the polygenic trait, intramuscular fat content, comprises host gut traits..."
- 6.Also, add "IMF" abbreviation in line 92.
- 7.Line 171- "alrs" needs to be italicized
- 8.Line 277-280: Were these findings also examined in the cecum?
- 9.Authors do an amazing job describing each MG abundance in detail. Good job linking the findings to relevant studies in the past.
- 10.Line 332- seems like the sentence is unfinished, "...encoding ATP-binding cassette transporters of antibiotic..."
- 11.Line 358-360: Did authors perform 16S rRNA sequencing to determine microbial composition?
- 12.Did the authors examine sex differences in relation to microbial genes?

Reviewer #1

1. All the results about the associations of the changes of MGs abundances in cecum lumen with IMF were inferred from association study. There were no any experiments to confirm these hypotheses.

ANSWER: No other experiments are cited in the paper, but we have a metabolomic study of the caecal content in progress which will cast light on this point, as it appears in some early analyses presented in the next EAAP congress in Davos (Zubiri-Gaitán A., Blasco A., Hernández P. 2021. The microbial metabolome and its role in intramuscular fat deposition. 72th Annual Meeting of the European Federation of Animal Science. Davos, Switzerland. 30 August – 03 September 2021).

2. The analyses in this manuscript only focused on microbial genes. It should be interesting that the authors can link the microbial genes affected by selection for IMF to their corresponding bacterial species, and then, discussed the effects of these bacterial species on host IMF and the possible mechanism of these bacterial species influencing IMF through MGs.

ANSWER: Our study includes the identification of those microbial taxa carrying the highest number of the 122 identified microbial genes in their genome (see lines 201-213 of the revised manuscript, Figure 3 and Supplementary Table 5). Whilst our results indicated Akkermansia genera and other microbes identified at upper taxonomic levels (e.g. Firmicutes phyla or Clostridiales order) as the main carriers of the identified microbial genes, we agree with the reviewer that a further comprehensive study of the microbial taxa host-genetically linked to IMF would complement our results. As we collected the whole metagenome sequencing data from the rabbit cecum samples, we are extending our cecum microbiome identification to taxonomical analysis **in coming research**.

3. The authors should provide the evidences that the genotypes associated with IMF had already fixed in two divergent lines, but not still be segregated.

ANSWER: We do not expect to have fixed genes in both lines but increasing the frequencies of the genes involved in intramuscular and body fat in one line and decreasing their frequencies in the other line, since the traits depend on a large number of genes, and selection is performed based in the phenotype of intramuscular fat. Some of the genes may have been fixed, but this is largely irrelevant for the purpose of the paper, since having substantial differences between lines in their incidence is enough to detect genes involved in fat deposition.

4. line 48: ".....are restricted to phenotypic studies.....",
However, phenotypic values are significantly correlated with genetics.

ANSWER: Of course, phenotypic values are related to with genetic values, otherwise association studies will be useless. However this correlation is not 1, which means that extreme phenotypes are likely to be extreme due to environmental effects. Microbiome composition is influenced by many environmental effects as diet, housing conditions, exercise rates, and a large amount of non-identified environmental causes. Our experimental design intends to capture the genetic causes by divergent selection on IMF, so that the differences between populations are due only to genetic causes. Nevertheless, in a selection process some differences can appear just by chance, by sampling genes from generation to generation (what is known as genetic drift), therefore we have selected only genes that were common to both procedures, phenotypic association (**PLS**) and genetic different between divergent lines (**DA-PLS**).

5. Line 81-88: this paragraph should be moved to the discussion section of methods section.

ANSWER: We find the paragraph necessary in order to introduce one of the aims of the study, which is to apply an additive log-ratio transformation to microbiome data as an alternative to the more common centered or isometric log-ratios. We are interested in stressing that our ALR procedure (selecting as reference gene the less variable one) has the same statistical good properties as the other methods but much simpler interpretation. The topic is further developed in methods, and we have included a note about it in the discussion, following the referee recommendation (lines 118-120).

6. Line 90: in this manuscript, the authors did not perform the metabolomics analysis, so it should be the functional capacities of cecum microbiome based on metagenomic sequencing data.

ANSWER: Thanks for the suggestion; we have modified the text accordingly.

7. In my opinion, the authors should measure the concentrations of serum LPS, peptidoglycans, N-acetylneuraminic acid, lipoproteins, and other metabolites that the authors emphatically discussed in the result section of the manuscript. It would help to confirm that the changes of cecum microbiome functional capacities indeed led to the shifts of the corresponding metabolites, and these metabolites finally resulted in the different IMF contents.

ANSWER: We agree with the reviewer. As previously mentioned, we have a metabolomics study of the caecal content in progress which we expect

will cast light on this point.

8. Line 250-254: However, many studies have suggested that *Akkermansia* spp. can alleviate metabolic syndrome features in obese human. e.g., 10.1038/s41591-019-0495-2; 10.1038/s41591-019-0516-1.

ANSWER: Many thanks for the comment; we have included a note about it in lines 211-213 of the revised manuscript.

9. line 369-370: Many studies in other mammals (such as pigs) have suggested different genetics backgrounds for IMF and body fat, e.g., backfat.

ANSWER: A large part of the genetic background of IMF and body fat is different, but part of it is common as clearly evidenced by the correlated response observed in body fat depots when divergently selecting by IMF during 10 generations. The difference in perirenal fat weight, which is the main fat deposit in rabbits, was 5.11 g in generation 10 (1.36 SD), or in body fat % was 1.5SD, see Supplementary Table 1 and lines 322-324). A correlated response in body fat depots when selecting for IMF has been also evidenced in other species (e.g. pigs (Schwab et al. 2006 *J. Anim. Sci.* 87:2774-2780) or cattle (Saap et al. 2002 *J. Anim. Sci.* 80:2017-2022).

Reviewer #2 (Remarks to the Author):

The manuscript entitled "Comprehensive comparison of the cecum microbiome functional core in genetically obese and lean hosts under the same environmental conditions" is interesting, but the current form is not acceptable, it needs to be improved by addressing several below issues:

1. From what I know, there are some studies in the world which research about it. So, is this new topic? Authors should highlight more the reason why they did this work.

ANSWER: As we say in the paper, obesity comprises increased rates of lipid deposition in body fat and also in intramuscular fat depots. Whereas the role of the microbiome composition in the former has been extensively studied, there are much less published works about the microbiome influence on IMF, and they are restricted to phenotypic studies in livestock species. Phenotypic values are related to with genetic values, but this correlation is not 1, which means that extreme phenotypes are likely to be extreme due to environmental effects. Our experimental design intends to capture the genetic causes by divergent selection on IMF, so that the differences between populations are due only to genetic causes. Nevertheless, in a

selection process some differences can appear just by chance, by sampling genes from generation to generation (what is known as genetic drift), therefore we have selected only genes that were common to both procedures, phenotypic association and genetic different between divergent lines. This is the first original characteristic of our work. Moreover, we have analysed the data taking into account its compositional nature (many research works have missed this point), but using an original procedure: selecting as reference gene the less variable one and analyse the data using ALR technique after proving that this is as correct as the more common CLR and ILR procedures.

2. In the Main text, the consequences of obesity should be more clarified which will support the reasons of studies on obesity in general and on this topic in specific, several importance references you should read such as: (1). Medical Consequences of Obesity, The Journal of Clinical Endocrinology & Metabolism, Volume 89, Issue 6, 1 June 2004; (2), An update on obesity: Mental consequences and psychological interventions, Diabetes Metab Syndr. 2019 Jan-Feb;13(1):155-160; and (3).An update on physical health and economic consequences of overweight and obesity. Diabetes Metab Syndr. 2018 Nov;12(6):1095-1100.

ANSWER: We appreciate this comment, which improves the quality of our work. We have included a sentence accordingly in the main text (lines 28-29), quoting the papers suggested.

3. In the second main text paragraph, you mentioned that “Rabbits are a more appropriate experimental model than the most widely used mice for the study of lipid metabolism in humans, as their lipoprotein metabolism, cardiovascular system and obesity-related clinical signs are more similar to those of human than the same systems of mice”. As far as I know, in previous studies, researchers still used mice as their research subject. Why is there a difference in the study subjects between the studies?

ANSWER: We work with rabbits because they are a wonderful model for biological research. In reproduction they offer several advantages: as ovulation is induced by coitus we know when fecundation takes place, so we can do accurate research involving the age of the embryos; moreover, implantation sites can be counted by laparoscopy without affecting litter size. In research involving intestinal tract content we have the peculiarity of caecotrophy: rabbits protect their faeces by a mucous protein envelope (soft faeces) and ingest them directly from the anus, producing later the common hard faeces; we are examining the possibilities of caecotrophy for microbiome research. Moreover, as we say in the Main text section, lipoprotein metabolism, cardiovascular system and obesity-related clinical signs are more similar to those of human than the same systems of mice. Rabbits are easier to manage for clinical operations like ovariectomy, embryo

recovering, embryo transfer etc, and embryos are easy to vitrify. Rabbit meat is consumed in many countries, particularly in the south of Europe, whereas it is very rare that rats are eaten (they are consumed in places like Senegal and they were until recently in Valencia, Spain, for the popular dish “paella”), thus rabbits can be used in studies of meat quality, since carcasses are cheap and easy to dissect. Rabbits have a short generation interval (six months) which make them convenient for genetic studies. We understand that rabbit facilities are more expensive than mice facilities, but rabbits offer many more possibilities for research.

4. In the main text and introductory paragraph, you should give some information about the microbiome in order to see overview of the whole article.

ANSWER: We have followed the reviewer’s suggestion (see line 11). Information about the microbiome in the main text can be found in lines 87-91.

5. About the layout of the article, I find it not reasonable. If I were you, I would write the research method first, then the results and discussion.

ANSWER: This is a matter of opinion; we have followed the instructions of the “Style and formatting guide” of Communications Biology.

6. In the Animal method, you mentioned in detail about the research subject, rearing process and amount of food. But, I haven't seen you mention farming conditions. such as laboratory conditions, etc..

ANSWER: We have extended the housing description in methods (lines 390-394). Cecum samples are collected in the slaughterhouse (added, line 406).

7. In the Animal method, you mentioned that “A divergent selection experiment for intramuscular fat content in rabbits was performed during 10 generations” Why do you use 10 generations of mice for research?

ANSWER: I presume you mean 10 generations of rabbit for research, since I answered the mice question in (3). We decided to select for IMF until both lines were not overlapping (Suppl. Fig. 2), which happened in generation 10. As selection is divergent, this is equivalent to performing 20 generations of selection for IMF.

8. In the method, in a study on the same topic, they used SSU rRNA gene amplification and pyrosequencing methods to evaluate the influence of host genetics on bacterial community composition (Campbell, J., Foster, C., Vishnivetskaya, T. et al, 2012), but your study does not cover this approach. Why?

ANSWER: Thanks for the interesting paper of Campbell et al. The recent research of the first author of this paper suggests a stronger host-genome influence on the functional microbiome (microbial genes) than on the microbial taxonomic composition identified at genera level in rumen (<https://www.researchsquare.com/article/rs-290150/v1>), and a stronger host-genomic association of the former with traits of interest. This could be explained by microbial genes being involved in producing specific substrates or mediating a specific pathway that interferes with the trait, whilst each microbial genus expresses many microbial gene functions. Another hypothesis is the functional versatility within different species or clades classified in the same genus, different niche specificity; or due to horizontal transfer of genes among microbial species. In our study we are focused on the microbial gene abundances influenced by the same host-genomic basis as IMF, but we also included the identification of those microbial taxa carrying the highest number of the 122 identified microbial genes in their genome (see lines 201-213 of the revised manuscript, Figure 3 and Supplementary Table 5). Nevertheless, we agree with the reviewer that a further comprehensive study of the microbial taxa host-genetically linked to IMF would complement our results. As we collected whole metagenome sequencing data from the rabbit cecum samples, we will extend our cecum microbiome identification to taxonomical analysis in further research.

9. How did you classified which rabbits were obese and which were not?

ANSWER: The rabbits from the obese line were selected for 10 generations for high IMF, whilst the non-obese were selected for 10 generations for low IMF. IMF is an obesity indicator (see Goodpaster et al., 2000 reference number 49), and the selection had a positive correlated response in lipid deposition in other muscles, body fat depots, liver size, lipogenic activity of liver, muscle and adipose tissue, etc. (see lines 76-81).

10. In the Microbial gene abundance measurements method, there are quite a few techniques applied, you should name each technique and talk in more detail. From there, readers can understand more easily

ANSWER: Thanks for the comment. The description of each technique has been extended (see lines 447-469).

11. In the result, you have only concluded about the function of the

cecum microbiome and their mechanisms, but the main objective of the study is the comparison of the cecum microbiome functional core in genetic obese and lean hosts under the same environmental conditions, you haven't mentioned it yet.

ANSWER: Thanks for the comment. We have now specified that we are studying the core microbiome it in the results section.

12. One of the most important thing is that the molecular mechanisms underlying the interactions of the cecum microbiome and the hosts observed in this studies have not been enough studied well here, or the ways authors presented and interpreted the data were not so good that makes the readers difficultly to get the right information

ANSWER: The molecular mechanisms explaining the interactions of the cecum microbiome and the host could be elucidated by GWAS studies aiming to identify the host genes underlying microbial gene functions. But this is out of the scope of this study, and larger datasets are required.

13. In this research, you just stopped at making the conclusions of the study, stating some scientific bases and conclusions of previous studies. But you have not mentioned the shortcomings and applicability of the research.

ANSWER: Many thanks for this point. We have added a sentence in the discussion clarifying the gap of knowledge that our study aims.

Reviewer #3 (Remarks to the Author):

Major Claims: Authors claim that their work elucidates that microbial biosynthesis of LPS, peptidoglycans, lipoproteins, mucin components, and NADH reductases are influenced by host genetic determination for lipid accretion in muscles. The authors also discuss the criterion to select adequate sets of additive log-ratios for compositional analysis. This emphasizes the importance of compositional transformation when performing covariance-based analysis. These claims are novel and would be of interest to the community.

Statistical Analysis: Authors wrote a very detailed description of the statistical analyses completed for the project. All statistical measures appear to be appropriate and reproducible.

1. Line 32-35= clarification is needed, do authors mean that the gut microbiome is determined by host genes along with other factors? If so, the sentence could be rephrased.

ANSWER: We mean microbiome is influenced by the genes of the host that control the pH, motility, immune response of presence of specific receptors for metabolites with microbial origin. We have re-phased the sentence.

2.Lines 45-46; the definition of “obesity” and its role in lipid deposition could be clear.

ANSWER: We have modified the sentence, including the paper of Bay et al., where obesity is specified to comprise two factors: increased mass of adipose tissue and enlarged fat cells.

3. Can the authors please explain why the cecal contents were examined?

ANSWER: We did a preliminary study evaluating the correlated responses to selection for IMF on ileum, cecum and faeces (soft and hard) functional microbiome, and obtained similar results in the three sites (Martínez-Alvaro et al. 37th ISAG Conference 7-12 July 2019, Lleida, Spain). For practical reasons, we selected caecal microbiome as a standard. However, we are conscious about the advantages of using faeces as microbiome samples in genetic selection studies and we have some projects associated to this idea, taking into account that rabbits have two kinds of faeces, soft and hard, the former ones have a mucus protein protection and are ingested by rabbits directly from the anus. Colon microbiome was not evaluated.

4. Were fecal contents in the small intestine or colon examined? Why not pellets?

ANSWER: Please see ANSWER above.

5. Lines 91-93; Hypothesis needs to be moved to the introduction. Also, the sentence structure makes the hypothesis difficult to read. Maybe format as the following, “The main hypothesis of this study is that genetic determination of the polygenic trait, intramuscular fat content, comprises host gut traits...”.

ANSWER: Thanks for the comment. The hypothesis is already formulated in the introduction, we repeat it at the start of the results in order to help the reader matching our hypothesis with our results.

6.Also, add “IMF” abbreviation in line 92.

ANSWER: We have not abbreviated IMF in the text, but only when strictly necessary in figure 1, to make the document for readable.

7.Line 171 “alrs” needs to be italicized

ANSWER: Done

8.Line 277-280: Were these findings also examined in the cecum?

ANSWER: The three studies quoted (Million et al., Schwiertz et al. and Luo et al.) analysed stools/faecal samples in humans/pigs. We have specified it in the text, many thanks for the comment!

9.Authors do an amazing job describing each MG abundance in detail. Good job linking the findings to relevant studies in the past.

ANSWER: Many thanks!

10.Line 332- seems like the sentence is unfinished “...encoding ATP-binding cassette transporters of antibiotic...”

ANSWER: Thanks for the comment. We have amended it.

11.Line 358-360: Did authors perform 16S rRNA sequencing to determine microbial composition?

ANSWER: No, we did not. However, as we collected whole metagenome sequencing data from the rabbit cecum samples, we will extend our cecum microbiome identification to taxonomical analysis in further research.

12. Did the authors examine sex differences in relation to microbial genes?

ANSWER: At 9 weeks of age, rabbits do not show sexual dimorphism in fat traits. To satisfy reviewer’s suggestion, we have evaluated the effect of sex in the functional microbiome in a PERMANOVA analysis with 999 permutations. Sex explained a negligible proportion of the total variance inherent in the microbial genes database (1.2%, P-value =0.311).

REVIEWERS' COMMENTS:

Reviewer #2 (Remarks to the Author):

The authors have responded to my comments and improved the manuscript